# Experimental Study on the Physicochemical Properties of Asphalt Modified by Different Anti-Stripping Agents and Their Moisture Susceptibility with Aggregates

**DOI:** 10.3390/ma16134545

**Published:** 2023-06-23

**Authors:** Ziyu Lu, Anqi Chen, Shaopeng Wu, Yuanyuan Li, Yingxue Zou, Yunsheng Zhu, Kaifeng Wang

**Affiliations:** 1School of Transportation and Logistics Engineering, Wuhan University of Technology, Wuhan 430063, China; 2State Key Laboratory of Silicate Materials for Architectures, Wuhan University of Technology, Luoshi Road 122, Wuhan 430070, China; 3School of Civil Engineering and Architecture, Wuhan Institute of Technology, Wuhan 430205, China

**Keywords:** asphalt, anti-stripping agent, moisture susceptibility, rheological properties, surface free energy, digital image processing

## Abstract

Erosion and the stripping effect of moisture on asphalt mixtures is one of the main reasons for the shortened service life of asphalt pavements. The common mean of preventing asphalt pavements from being damaged by moisture is adding anti-stripping agents (ASAs) to asphalt mixtures. However, the effect regularity and mechanism of anti-stripping agents on the physicochemical properties of asphalt is not exactly defined. This study compared the physical properties of ASA-modified asphalt (AMAs) to determine the optimal dosage and investigated the rheological and adhesion properties. Based on the roller bottle method and water immersion method, the moisture susceptibility of AMAs with three particle sizes was investigated. The results showed that the modification of asphalt using anti-stripping agents was a physical modification. At the optimum dosage of anti-stripping agents (0.3%), the basic physical properties of AMA1 were the most desirable. ASA2 increased the resistance of asphalt for deformation at high temperature by 46%, and AMA3 had the best low-temperature performance. ASAs enhanced the dispersed and polar components in the asphalt binder, improving the adhesion energy of asphalt. AMA3 had the strongest adhesion to the aggregate, with an increase in adhesion work by 2.8 times and a 45% of increase in ER value. This was attributed to ASA3 containing with a large number of metal cations and polar functional groups. It was shown that ASAs provided the most improvement in the anti-stripping performance of asphalt mixtures with 9.5–13.2 mm particles. The amide ASA, phosphate ASA and aliphatic amine ASA improved the water damage resistance of asphalt by 65%, 45% and 78%, respectively. This study can help engineers realize the effects of different types of ASAs on the physicochemical properties of asphalt and select the most suitable type of ASAs according to the service requirements.

## 1. Introduction

Asphalt mixtures are exposed to moisture and atmospheric rainfall for long periods, and their interaction with water leads to bond failure, which in turn causes the asphalt mixture to be stripped [1]. Furthermore, the water will also weaken the adhesion of the pavement layer with the bedding layer and the base layer, weakening the load transfer characteristics of the entire road base pavement system, thus leading to stripping, loosening, holes, potholes, and other diseases. This process is known as asphalt pavement moisture damage [2,3]. Some of the factors that result in the stripping of asphalt mixtures are the material characteristics [4], mixture voids and connectivity [5], and hydraulic washout [6]. In general, the stripping damage of asphalt mixtures caused by water is mainly divided into two cases: adhesion damage of the asphalt–aggregate interface and internal cohesion damage of asphalt. The phenomenon of moisture damage seriously impairs pavement service life and causes a great loss of social and economic resources. Therefore, to extend the life cycle of pavements, it is necessary to reduce the moisture susceptibility of asphalt.

Aggregate type is associated with asphalt mixture wettability and contributes critically to the asphalt adhesion characteristics. Asphaltene is an asphalt component that plays a key role in the asphalt–aggregate interface performance, as it interacts with the mineral components in the aggregate to varying degrees: MgO > CaO > Al_2_O_3_ > SiO_2_ [7]. Aggregates with high SiO_2_ content (e.g., granite, sandstone) have worse adhesion properties than those with low SiO_2_ content (e.g., basalt, limestone). This is attributed to the weak bonding ability of the anionic component present in asphalt with siliceous aggregates, which can be easily replaced by water in combination with the aggregates [8]. Under water-filled conditions, for improving the stripping resistance of asphalt mixtures, inorganic mineral powders are usually used as fillers or additives, such as slaked lime, cement, and nano-clay, as a result of their low cost and good performance. However, inorganic mineral powder in the actual engineering application process is complex since the effect is not stable. Moreover, the asphalt agglomeration phenomenon also often occurs, affecting the rheological properties of the asphalt, causing it to be no longer suitable for improving the anti-stripping performance in a large number of asphalt mixtures. In addition, other researchers have found that polymers, such as polyethylene [9] and polyvinyl chloride [10], may also be beneficial in improving the water resistance of asphalt mixtures. However, they have not been used as the main means to improve the moisture susceptibility of asphalt because of their weak anti-stripping performance and the effect on the other properties of asphalt binders.

New liquid ASAs have been widely used in pavement engineering in recent years because of their small dosage, low cost, ease of construction, and high-temperature stability. These materials are organic polymer compounds, which are mainly divided into amine surfactants and non-amine surfactants [11], and the most commonly used are amides, fatty amines, phosphate esters, and phosphorus hydroxyl ASAs. There are polar and nonpolar groups in ASAs, which have a strong affinity for aggregate surfaces and asphalt, respectively. The positively charged polar groups in ASAs are oppositely attracted to the negative charge of the aggregate surface, providing a strong adsorption effect. It is usually used at 0.25–0.75% of asphalt binder weight and can be added to both molten asphalt binder and aggregate [12]. Joel R.M. et al. incorporated a warm-mix asphalt surfactant (the main component is sodium aluminosilicate) to lower the temperature in a rubberized asphalt mixture production and found the strength of wet specimens to be significantly improved by indirect tensile strength tests. This indicates that the surfactant has the potential to decrease moisture damage in asphalt mixtures. Moreover, the surfactant had essentially little influence on the physical and rheological properties of the asphalt in the range of 0.1–0.5% of the used admixture [13]. Shu et al. added amide and aliphatic amine ASAs to asphalt, investigated their effects on the adhesion properties, and observed that both were effective in improving the asphalt–aggregate adhesion work. The freeze–thaw splitting results showed that the ASAs were able to increase the tensile strength ratio (TSR) of the asphalt after aging from less than 70% to more than 80% [14]. Syed Ashik et al. characterized the water-induced failure of various amino-based modifiers on asphalt mixtures using contact angle tests and semicircular bending tests, and the control group containing amine-based additives exhibited a higher average fracture energy and reduced moisture susceptibility of the mixture, as in accordance to those of the surface free energy tests [15]. Afshar A. et al. incorporated additives, such as amide-based surfactants, into the warm-mix asphalt for recycled asphalt pavement and found that the water susceptibility and rutting resistance of asphalt mixtures were improved [16]. However, the organic polymer ASA provided an efficient mean to solve the problem of moisture damage, the effect of the chemical composition in different anti-stripping agents on the physicochemical properties of asphalt is not clear. Additionally, the modification effect of anti-stripping agents on the moisture sensitivity of asphalt mixtures under different conditions is pending for study [17]. In addition, most studies have not extensively used multiple ASAs to compare performance test results, and the effect on the physical, chemical, and rheological properties of asphalt by various ASAs have not been made clear [18]. These are the key issues that need to be explored to further optimize the formulation of ASA to avoid the occurrence of moisture damage in asphalt mixtures and to ensure the overall performance of the pavement.

Therefore, amide, phosphate ester, and fatty amine ASAs were selected in this study, and the chemical composition of these three ASAs was investigated using X-ray fluorescence spectroscopy (XRF) and Fourier transform infrared spectrometer (FTIR). The physical properties of asphalt, including the softening point, penetration, and ductility, were used to determine the optimum dosage of ASAs. By using dynamic shear rheometer (DSR) and bending beam rheometer (BBR), the rheological properties of AMAs at high and low temperatures were evaluated. Based on the surface free energy principle, the enhancement mechanism on the anti-stripping performance of asphalt was analyzed. The moisture susceptibility tests (rolling bottle method and water immersion method) of the asphalt mixture were combined with digital image processing to assess the performance of AMA under dynamic water and high-temperature conditions, respectively.

## 2. Materials and Methods

### 2.1. Raw Materials

#### 2.1.1. Asphalt Binder and Aggregates

The material used was an asphalt binder from Ezhou Kepai Asphaltic Products Co., Ltd. (Ezhou, China) with Pen 60/80 (abbreviated as AB). Table 1 provided details about the physical properties of the AB. Basalt aggregates with a particle size of 4.75–9.5 mm, 9.5–13.2 mm, and 13.2–16 mm were collected from Jingshan, Hubei Province. Their properties indexes were shown in Table 2.

#### 2.1.2. Anti-Stripping Agents

Three commercial anti-stripping agents (ASAs) with different major components were used to modify the asphalt binder properties. ASA1, ASA2, and ASA3 refer to amide, phosphate ester, and aliphatic amine ASAs, respectively. The characteristics of ASAs were listed in Table 3.

### 2.2. Experimental Methods

Figure 1 illustrates the research program on the performance of asphalt modified with different anti-stripping agents and their moisture susceptibility with aggregates. 

#### 2.2.1. Preparation of AMAs

The melt blending method was used to prepare the AMA. According to the manufacturer’s recommendation, the four dosages of ASAs in the asphalt were set at 0.2%, 0.3%, 0.4%, and 0.5%, respectively. The asphalt binder was heated to a flowing state at 140 °C. Thereafter, ASA was slowly dripped into the asphalt binder through a dropper with a rubber tip and stirred at 1000 r/min. At a constant temperature of 140 °C, the ASA and asphalt binder were mixed by a high-speed shearing machine at 4000 r/min for 30 min. To control the experimental variables, a blank sample was created using the same procedure for the base asphalt binder.

#### 2.2.2. X-ray Fluorescence Spectroscopy Test

Powdered basaltic aggregates and three ASAs samples were subjected to qualitative and semi-quantitative chemical analysis using Zetium XRF spectrometer (PANalytical B.V., Almelo, The Netherlands). The test was performed by the pressed slice method, which is based on the principle of X-ray fluorescence spectroscopy. The elements were determined from F to U by also using OMNIC 9.2 standardless analysis software, and the content was determined in the identification range from ppm to 100%. The test results are presented in oxide mode.

#### 2.2.3. Fourier Transform Infrared Spectrometer Test

FTIR is a molecular functional group identification method that is commonly used to identify organic compounds, such as asphalt [21]. The mechanism of asphalt modification by ASAs was investigated using FTIR. The AB and AMAs were prepared using the potassium bromide compression method. In the wave number range of 4000–400 cm^−1^, the infrared spectra of the samples were obtained using Nicolet iS5 (Thermo Fisher Scientific Inc., Waltham, MA, USA) with the parameters of scan number and resolution setting at 32 and 4 cm^−1^, respectively. After that, the position, shape, and width of the peaks of the infrared spectra were observed in conjunction with Omnic 9.2 software for functional group analysis.

#### 2.2.4. Static Contact Angle Measurement and SFE Calculation

The sessile drop method is widely used by researchers as an SFE test method because of its simplicity and the reliability of the results [22]. On the assumption that the droplet shape on the smooth horizontal specimen surface only depends on surface tension and gravity, the measurement of the static contact angle for the asphalt and aggregate was carried out using the sessile drop method [21]. A total of 2 g of asphalt was weighed and heated to a liquid state and spread on the slide to form a smooth horizontal surface. After it cooled down, the asphalt contact angle sample was obtained. Additionally, the basalt aggregate was cut into 20 mm × 20 mm × 5 mm slices, and then the test surface was polished smooth using a polishing machine. An optical contact angle meter (OCA20, Dataphysics, Stuttgart, Germany) was used to obtain the contact angles of the samples. Four probe liquids (water, formamide, glycerol, and diiodomethane) with known SFE component parameters were used [23]. Of these, water, glycerol, and formamide were used to test asphalt samples, and water, formamide, and diiodomethane were used to test aggregate samples. In addition, when conducting the tests, each probe liquid was repeated three times.

Figure 2 depicted the calculation method of contact angle. Once the wetting process between the probe fluid and the solid surface reaches a steady state, the interfacial parameters of both the solid and the liquid satisfy the Young–Dupre equation [24]:
(1)$$\gamma_{s}-\gamma_{sl}=\gamma_{l}\cos\theta$$
where $\gamma_{s}$, $\gamma_{l}$, and $\gamma_{sl}$ are the SFE of the solid-phase and liquid-phase, and the solid–liquid interface, respectively. The θ is the contact angle between the interface.

Based on Fowkes’s theory [25], the SFE of a substance consists of polar and dispersive components for either liquids or solids, as shown in Equation (2).
(2)$$\gamma =\gamma^{p}+\gamma^{d}$$
where $\gamma^{p}$, $\gamma^{d}$ express the polarity and dispersion components of the SFE, respectively.

The interface free energy of the solid and liquid is expressed in Equation (3) [26,27].
(3)$$\gamma_{sl}=\gamma_{s}+\gamma_{l}-2\sqrt{\gamma_{s}^{p}\gamma_{l}^{p}}-2\sqrt{\gamma_{s}^{d}\gamma_{l}^{d}}$$
where $\gamma_{s}^{p}$, $\gamma_{l}^{p}$ represent polarity components for the solid-phase and liquid-phase, respectively, and $\gamma_{s}^{d}$, $\gamma_{l}^{d}$ represent dispersion components for the solid-phase and liquid-phase, respectively.

Following that, Equation (4) is established from Equations (1) and (3) to calculate the SFE components of the solid-phase [28]. Three equations are established through the SFE components of the three probe fluids, considering the solid surface energy parameters as coefficients and the probe fluid surface energy parameters as independent and dependent variables. By using linear analysis, the polar and dispersive parameters of the solid phase were discovered. The SFE components for the four probe liquids acquired from the literature were displayed in Table 4 [23].
(4)$$\frac{\left(1+\cos\theta \right)}{2}\frac{\gamma_{l}}{\sqrt{\gamma_{l}^{d}}}=\sqrt{\gamma_{s}^{p}}\sqrt{\frac{\gamma_{l}^{p}}{\gamma_{l}^{d}}}+\sqrt{\gamma_{s}^{d}}$$

In addition, the effect of ASA was quantified by calculating the adhesion energy and the stripping energy. When THE asphalt binder is mixed with the aggregate, the energy released by the formation of the bonding interface is adhesion energy. Its calculation formula is shown in Equation (5) [29,30]. The release of energy when two asphalt molecular surfaces fuse to form an interface is defined as cohesive energy, which was calculated as shown in Equation (6) [31]. The asphalt–aggregate interface breaks under the effect of water replacement, and the whole system forms two new interfaces, asphalt–water and water–aggregate, and the amount of change in the surface free energy during this process is defined as the stripping work. A higher stripping energy means that the asphalt on the aggregate surface is more likely to strip, the calculation of which is shown in Equation (7) [32].
(5)$$W_{ad}=2\sqrt{\gamma_{a}^{p}\gamma_{s}^{p}}+2\sqrt{\gamma_{a}^{d}\gamma_{s}^{d}}$$
(6)$$W_{co}=2\gamma_{a}$$
(7)$$W_{st}=2\gamma_{w}+2\left(\sqrt{\gamma_{a}^{p}\gamma_{s}^{p}}+\sqrt{\gamma_{a}^{d}\gamma_{s}^{d}}\right)-2\left(\sqrt{\gamma_{w}^{p}\gamma_{a}^{p}+\gamma_{w}^{d}\gamma_{a}^{d}}\right)-2\left(\sqrt{\gamma_{w}^{p}\gamma_{s}^{p}+\gamma_{w}^{d}\gamma_{s}^{d}}\right)$$
where $W_{ad}$, $W_{co}$, and $W_{st}$ are the adhesion work, cohesion work, and stripping work, respectively; $\gamma_{a}^{p}$ and $\gamma_{s}^{p}$ represent the polarity component of asphalt and aggregate, respectively; and $\gamma_{a}^{d}$ and $\gamma_{s}^{d}$ represent the dispersive component of asphalt and aggregate, respectively.

The adhesion-stripping energy ratio (ER) was used to visually characterize the surface energy differences of different asphalt samples. The $R_{as}$ was positively correlated with the stripping resistance of the asphalt film. Their calculation equations are shown in Equation (8).
(8)$$\mathrm{ER}=\frac{W_{ad}}{W_{st}}$$

#### 2.2.5. Dynamic Shear Rheometer Test

The rheological properties of asphalt materials vary significantly under different loads and temperatures when they are in the linear viscoelastic range, and the rheological properties of asphalt binders at high temperatures can provide a rough performance indication in the production and usage of asphalt binders [33]. A DSR test was performed in high-temperature sweep mode with a temperature rise rate set at 2 °C/min. At temperatures between 52–82 °C, various asphalt binders were loaded at 10 rad/s, always maintaining a strain of 12%. The rotor diameter and plate gap were 25 mm and 1 mm, respectively. The complex shear modulus (G*), phase angle (δ), and rutting factor (G*/sin δ) reflect the viscoelastic properties and deformation resistance of asphalt binders under high temperature, respectively [34].

#### 2.2.6. Bending Beam Rheometer Test

The cracking resistance of asphalt at low temperatures was characterized by the BBR test, which tests the flexural creep modulus of stiffness (S) and the slope of the double logarithmic curve of S versus the load action time (m-value) of the matrix asphalt and the three AMAs, as shown in Equation (1) [35,36]. The tests were carried out at −6, −12, and −18 °C. At low temperatures, the asphalt deformation resistance is measured by the S-value, and the creep relaxation capacity is characterized by the m-value. The S- and m-value of the asphalt are calculated in terms of the corresponding deformation values at 60 s [37]. For the reliability of the experimental results, five parallel tests were conducted for each sample.

#### 2.2.7. Rolling Bottle Test

A modified rolling bottle test (RBT) based on BS EN 12697-11 [38] was developed [39]. The adhesion and cohesion of the asphalt binders were characterized by measuring the degree of asphalt coating after a period of mechanical mixing force applied to the uncompacted asphalt mixture in a water environment. Samples for the RBT test were produced by mixing different asphalt binders with three different particle sizes of aggregates. The mixing temperature was maintained at 140 °C, and 40 g of aggregate samples were uniformly coated with 1.4 g of asphalt binder. The loose mixture was weighed after being cooled at room temperature for an hour. Before the test, each test bottle was filled with pure water to half of its volume, and then the mixture was slowly placed and sealed in the bottle. The test bottle was rotated on the drum at 15 rad/min, and the ambient temperature was kept at 20 °C for about 24 h [40]. Finally, the loose mixture in the test bottle was removed to be evaluated for stripping using the method that will be discussed in Section 2.2.9. Additionally, five replicates of each sample were prepared.

#### 2.2.8. High-Temperature Water Immersion Test

Under the combined effect of high temperature and water conditions, the stripping resistance of the AMA was evaluated using an improved high-temperature water immersion test [41]. First, 3.5 g of molten asphalt binder was uniformly wrapped with 100 g of aggregate at 140 °C. Five sets of parallel samples were prepared. Next, the loose mixture was soaked for 30 min in 80 °C water. After the water immersion cycle, the loose mixture was removed to cool to room temperature and was evaluated for stripping by the method presented in Section 2.2.9.

#### 2.2.9. Evaluation Method of Anti-Stripping Performance

In the studies of moisture damage on asphalt mixtures, researchers usually use a visual method to evaluate the asphalt stripping rate, which is subject to human factors and excessive errors [42]. By introducing two indicators, mass stripping rate and area stripping rate, the degree of adhesion damage of the asphalt binders was evaluated integrally. The asphalt binder flaked off the aggregates to different degrees before and after the rolling bottle and water immersion tests. The mass stripping rate is a measure of the change in the mass of the loose mixture, which in turn characterizes the degree of total asphalt stripping. The mass stripping rate ($R_{MS}$) of the loose mixture was calculated by Equation (9) [43]:
(9)$$R_{MS}=\frac{M_{c}-M_{0}}{M_{c}-M_{s}}\times 100\%$$
where $M_{c}$ represent the mass of the loose mixture, $M_{0}$ represent the mass of aggregate, and $M_{s}$ represent the mass of the loose mixture after the test.

The area stripping rate is a measure of the change in the asphalt coating area on the loose mixture and characterizes the asphalt–aggregate interface adhesion. Images of the tested loose mixtures were taken in a well-lit environment and imported into Image J software to analyze the stripping area [41,44]. Figure 3 depicts the digital image processing. The full-field color values of the RGB photos were analyzed by Image J for binarization, and the threshold size of the grayscale image was adjusted so that the black and white areas were highly matched to the actual asphalt coating areas. The area of white pixel area was then counted by the analysis tool of the software. The asphalt area stripping rate ($R_{AS}$) is the percentage of white pixel area to the aggregate area, as shown in Equation (10):
(10)$$R_{AS}=\frac{A_{s}}{A_{a}}\times 100\%$$
where $A_{s}$ and $A_{a}$ express the asphalt stripping area and the aggregate area, respectively.

## 3. Results and Discussion

### 3.1. Modification Mechanism of ASAs

#### 3.1.1. Elemental Composition of Aggregates and ASAs

The performance of asphalt mixtures is largely influenced by the chemical composition of the aggregates. The chemical composition of the ASA plays a crucial role in asphalt modification. Table 5 shows the XRF results for the aggregate and ASAs. Alkalinity (CaO/SiO_2_ content ratio) was used as a parameter for the aggregate affinity to asphalt. The basalt had a low alkalinity of 0.27, which indicated that the basalt aggregates are acidic and may have weak bonding to the asphalt in a moisture-heavy environment [41]. It is attributed to the siliceous minerals of the aggregate surface having a large quantity of hydroxyl groups, which form stronger hydrogen bonds with water than with asphalt. Therefore, water tends to replace the asphalt at the asphalt–aggregate interface [45]. The ionic bonding interaction between the polar cationic ends in ASAs and the aggregate surface has a strong anti-stripping effect. ASA1 contains a variety of metallic and non-metallic elements, with high contents of S, Fe, and Cr at 19.4%, 24.5%, and 7.9%, respectively. It is shown that the high content of these metallic elements helps to inhibit moisture damage in asphalt [46]. ASA2 mainly consisted of non-metallic elements, with 57.8% of S elements and 27.2% of P atoms. The main components of ASA3 are metallic elements. The enhancement of polar interactions between asphalt and aggregate by these metal ions and heteroatoms is the reason for the improved interfacial adhesion.

#### 3.1.2. Chemical Components of ASAs

Figure 4 presented the FTIR spectra of the AB and AMAs. In comparing the IR characteristic bands of the three ASAs separately in Figure 4, a total of six highly coincident characteristic peaks were found in the 2800–3000 cm^−1^, 1350–1450 cm^−1^, and 723 cm^−1^ wavenumber ranges. The characteristic peaks at 2920 cm^−1^, 2852 cm^−1^, and 1461 cm^−1^ were attributed to the occurrence of the asymmetric and symmetric stretching and vibration as well as the in-plane bending vibration of C−H in methylene, respectively. The asymmetric stretching vibration and symmetric bending vibration of C−H in methyl can be identified at 2959 cm^−1^ and 1373 cm^−1^, respectively. The methylene chain segment (−(CH_2_)n−) (n ≥ 4) at 723 cm^−1^ undergo synergistic motion. The result indicates that there may exist alkanes, aromatic hydrocarbons, and their derivatives among the three types of ASAs.

In addition, Table 6 displays the infrared characteristic peaks of ASAs in the 400–4000 cm^−1^ band. ASA1 belongs to the amide ASAs, and the vibration of C=O in tertiary amines corresponded to the peak centered at 1660 cm^−1^. The absorption peaks of both the amine groups and C=O in tertiary amide were weaker than in those with methylene, indicating the presence of a long alkyl chain of amide in ASA, which can have better compatibility with asphalt. The adsorption of polar groups in amide played a role in forming a high-strength bond in the asphalt–aggregate interface. Moreover, the remaining absorption peaks in ASA1 imply carboxylic acids, benzene rings, aromatic oxides, and aliphatic ethers might be present. ASA2 is a phosphate ester ASA, the absorption peaks of which are at 1200 cm^−1^, 1013 cm^−1,^ and 808 cm^−1^, corresponding to the infrared characteristic absorption peaks of phosphate esters. Further evidence that aromatic esters might be exist in ASA2 come from the absorption peaks at 3028, 1741, 1604, and 872 cm^−1^. ASA3 is an aliphatic secondary amine ASA, the characteristic functional groups of ASA3 corresponded to the peaks centered at 3310 cm^−1^, 1554 cm^−1,^ and 1095 cm^−1^. The presence of carboxylic acid and amide in ASA3 was indicated by the typical peaks for C=O at 1643 and 1719 cm^−1^. Additionally, the high-frequency peak bands at 1182–1315 cm^−1^ were attributed to the stretching vibration of C−N in the benzene ring. The stretching mode of the 1,4 para-substituted benzene ring was detected at 871 cm^−1^. Furthermore, ASA3 was shown to potentially contain carboxylic acids and amides. These polar groups contribute the most to the electrostatic interaction at the asphalt–aggregate interface and can prevent the fracture of the interface [47].

The infrared spectrogram of asphalt, which was modified with a large dosage of ASA was investigated to study the mutual reaction between the two substances. Compared to the FTIR spectra of AB, the FTIR spectra of the asphalt containing ASAs were more complex. In Figure 3, four absorption peaks belonging to asphalt spiked with 20% ASA1 were identified. The infrared characteristic peak of ν_as_ (P−O−C of phosphate ester) in the modified asphalt with the addition of ASA2 can be observed compared to AB. Moreover, the fatty secondary amine ASA is more similar to the asphalt composition, and only the stretching vibration of the para-substituted benzene ring can be observed, which can be better integrated with the asphalt. Overall, no additional absorption bands were found in the modified asphalt other than those already present in the asphalt binder and ASAs. This indicated that there is basically no chemical reaction within them.

### 3.2. Effect of ASAs on the Physical Properties of Asphalt

Figure 5 gives the physical properties of the asphalt added with different types and dosages of ASAs. With the increase in ASA dosage, the softening point of the AMAs increased and then decreased, the penetration first reduced and then improved, and the ductility continued to decrease. The ASAs hardly affected the asphalt properties at low dosages, but the heat oxidation aging in asphalt occurred during the high-speed shear process, causing a decrease in penetration and ductility and an increase in the softening point. The ASAs slightly impaired the temperature characteristics of asphalt. This mainly resulted from the benzene ring structure in the added ASAs, which enhanced the dispersing effect of asphaltene on the asphalt. As shown in Figure 5a, all three ASAs showed the maximum softening point at 0.2% dosage, and the softening point from small to large was AMA3 < AMA2 < AMA1, in which AMA1 increased by 1.4%. In Figure 5b, the AMAs showed the minimum penetration at 0.2% dosage, and the magnitude of their effects was consistent with the softening point results, among which AMA1 decreased by 3.1%. The ASAs brought down the viscosity of asphalt, the flowability was raised, and the high-temperature performance declined. Figure 5c showed that ASA1 had the least effect on asphalt ductility, followed by ASA2, and finally ASA3. To sum up, AMA1 had the best physical properties among the three AMAs, followed by AMA2. In addition, AMAs were prepared by selecting 0.3% as the dosage of the three ASAs to carry out subsequent tests to ensure better physical properties and an effective anti-stripping performance.

### 3.3. Effect of ASAs on the Rheological Properties of Asphalt

#### 3.3.1. High-Temperature Rheological Properties

ASA was added to asphalt as a modifier, which has a noteworthy impact on the rheological properties of asphalt [48]. DSR tests of three AMAs were employed to evaluate their rheological properties at a temperature range of 52–82 °C. Figure 6 presents the composite shear modulus (G*) and phase angle (δ) of the AMAs and AB. G* gradually decreased while δ slowly increased for the four asphalt samples as the temperature increased. This meant that the percentage of asphalt elastic component decreases, and the fatigue resistance of the material gradually deteriorates. Figure 7 displays the curves of the rutting factor with temperature for the different AMAs. The rutting factors of all three AMAs were larger than that of the AB: the rutting factor of AMA1 increased by 31%; the largest increase of 46% was observed for AMA2; and the AMA3 rutting factor had increased only by 8.5%. SHRP defines the temperature at $G^{*}/\sin\delta =1\;\mathrm{kPa}$ as the failure temperature. The failure temperatures of AB, AMA1, AMA2, and AMA3 are 67.28 °C, 68.83 °C, 69.76 °C and 67.63 °C, respectively, which meant that ASA2 visibly reduced the risk of permanent deformation for asphalt, and ASA1 had an improvement effect. Moreover, the high-temperature rheological properties of the asphalt were slightly affected by ASA3.

#### 3.3.2. Low-Temperature Rheological Properties

Figure 8 shows the S- and m-values of the AB and AMAs at low temperatures. The S of the four asphalts occurred to increase with decreasing temperature, and the m-values gradually decreased. In addition, the S-value was greater than 300 MPa and the m-value was less than 0.3 at −18 °C, which did not satisfy the SHRP [49]. This implied that the AB and AMAs would fail with the low-temperature performance at −18 °C. It was worth noting that the S- and m-values had different patterns of variation, so k = m/S was adopted as an indicator to comprehensively discuss the rheological properties of asphalt [50]. The k-value is proportional to the low-temperature characteristics of asphalt. The k-values of asphalt at −6 °C, −12 °C and −18 °C are shown in Figure 9. The k-values of the AB and AMAs show the same regularity at −6 °C and −12 °C: AMA3 > AB > AMA1 > AMA2. The regularity differs when the temperature reaches −18 °C because the temperature is lower than the failure temperature. Among them, the most significant difference in the k-value of each sample was observed at −6 °C. Compared with the AB, the k-value of AMA3 increased by 39%, while AMA1 and AMA2 decreased by 14% and 33%, respectively. Table 7 shows the failure temperature of the asphalt, which refers to the fitted temperature at 300 MPa and 0.3 after linear fitting of the S- and m-values of the asphalt. Lower failure temperatures would mean better low-temperature rheological properties. The failure temperatures of AMA1 and AMA2 were 4% and 11% higher than those of AB, respectively. However, ASA3 reduced the failure temperature of asphalt by 5%. Therefore, it can be determined that ASA1 and ASA2 reduce the low-temperature deformation resistance of asphalt, and ASA3 has an improvement effect on it.

### 3.4. Surface Free Energy Analysis

The modified asphalt was prepared based on the optimal dosage, and surface free energy was applied to analyze the mechanism of the improved adhesion properties at the asphalt–aggregate interface by the ASAs. Table 8 demonstrates the contact angles that the three probing liquids generated with the asphalt. Table 9 displays the surface energy components of the AB and AMAs. The polarity components of the AB and AMAs were significantly smaller than those of the dispersion components, indicating that the asphalt was a weakly polar material. After the addition of the ASAs, the dispersion components of the AMAs were all slightly increased. Moreover, the polarity components of AMA1, AMA2, and AMA3 increased to 2.4, 2.11, and 5.42 mJ/m^2^, respectively, and the polarity component had a higher incremental than the dispersion component [32]. It is worth mentioning that the polar component content of AMA3 is 2.8 times higher than that of the base asphalt. This indicated that the mechanism of action of the ASAs is to increase the polar component within the asphalt binder.

Figure 10 showed the adhesion energy ($W_{\mathrm{ad}}$) and cohesion energy ($W_{\mathrm{co}}$) of the AB and AMAs. Compared with the remaining AMAs, AMA3 showed the highest $W_{\mathrm{ad}}$, which was 20.7% higher than that of the AB, and $W_{\mathrm{co}}$ was higher by 35.7% to 36.02 mJ/m^2^. The results indicated that ASA3 could promote both the internal cohesive properties of asphalt and its interfacial adhesion with the aggregate. Figure 11 shows the stripping work and ER values of asphalt in a water environment. The stripping energy of the AMAs was smaller than that of the AB after the addition of the ASAs, which indicated that the ease of water-induced stripping at the asphalt–aggregate interface was reduced, and the moisture susceptibility of the mixture was improved. The stripping energy of AMA3 was 17.48 mJ/m^2^, which meant that AMA3 was less prone to stripping compared to AMA1 and AMA2. When comparing the ER values of the AB and AMAs, the ER values of AMA1 and AMA2 increased by 7% and 3%, respectively, and that of AMA3 increased by 45%. At the molecular level, this might be explained by the adhesion between the amine group of ASA3 and the silanol group on the surface of the aggregate. The silanol groups on the surface of the aggregate produced a stronger connection between the asphalt and the aggregate [18]. The SFE results reflected that the enhancement mechanism of the anti-stripping performance of ASAs lies in increasing the SFE of asphalt and the interfacial adhesion energy of the asphalt–aggregate interface, and decreasing the interfacial stripping energy, consequently improving the cohesion performance and adhesion performance of asphalt. The order of the enhancing impact of the three ASAs regarding the resistance to moisture damage of asphalt is AMA3 > AMA1 > AMA2.

### 3.5. Moisture Susceptibility of AMAs

#### 3.5.1. Effect of Dynamic Water on Anti-Stripping Performance at Room Temperature

Figure 12 shows the results of the asphalt stripping rate for different particle size aggregates under dynamic water conditions. The most severe stripping was observed for the mixtures prepared using aggregates with particle sizes from 9.5 to 13.2 mm in the dynamic water environment when the AB was used as the binder. The asphalt film mass loss rate of the mixture was as high as 25.4%, and the percentage of the asphalt stripping area was up to 39%. The addition of the ASAs markedly reduced these two indicators. This was most evident in AMA3, where the mass stripping rate and area stripping rate were reduced to 7.0% and 8.5%, respectively. Among the three AMAs, AMA2 had the highest asphalt stripping rate. Its asphalt mass stripping rate was still reduced by more than 10% compared to the AB, a trend that remained consistent with the results for the area stripping rate. However, ASAs showed more limited improvement in water damage resistance for 13.2–16 mm particle size mixes. For the 4.75–9.5 mm particle size asphalt mixtures, there was essentially no difference in the asphalt stripping rate between the AB and AMAs, which only fluctuated up and down by 3%. This indicated that ASAs did not have an improvement effect on the anti-stripping performance of 4.75–9.5 mm particle size asphalt mixtures under the dynamic water environment. Based on these facts, all three ASAs were shown to significantly enhance the anti-stripping performance of 9.5–13.2 mm particle size asphalt mixtures under the dynamic water environment and slightly improve the adhesion performance of 13.2–16 mm aggregates, and had less of an effect on the adhesion of 4.75–9.5 mm particle size aggregates. This might be accounted for by the smaller size of aggregates with a larger specific surface area and smaller ASA action area at the same dosage, which had insufficient improvement effect on the asphalt–aggregate interface. While the larger particle size of the aggregate had a larger individual mass, the aggregate between the mixture would produce higher frictional stress in the rolling bottle test. This effect on the anti-stripping performance of the mixture was greater than the effect of moisture stripping, so the anti-stripping performance of the AMAs was limited when it was used for large particle size aggregates. In addition, for the moisture susceptibility of 9.5–13.2 mm particle size aggregates, ASA3 showed the best improvement, followed by ASA1. The results were consistent with the surface free energy results.

#### 3.5.2. Effect of High-Temperature Water Immersion on Anti-Stripping Performance

Since the ASAs had the most significant effect on 9.5–13.2 mm particle size asphalt mixtures, 9.5–13.2 mm particle size aggregate was used as the research object to further determine the effect of ASAs on the stripping resistance of asphalt mixtures at high temperatures. Figure 13 demonstrates its asphalt stripping rate under high temperature and a hydrostatic environment. In comparison to base asphalt, the mass stripping rates of AMA1, AMA2, and AMA3 were decreased by 67.2%, 75.4%, and 78%, respectively, and their area stripping rates were all significantly reduced by about 79%. The reason for this phenomenon was considered to be that the mineral distribution on the aggregate surface, the microporosity distribution, and the angular characteristics determine the surface adhesion. After the addition of ASAs, the asphalt and aggregate form a tight bond at the interface through polar groups, and the adhesion no longer depended on the aggregate surface morphology [24]. As described above, the asphalt mixes prepared with all three AMAs were capable of reducing the stripping damage generated at the asphalt–aggregate interface and significantly reducing the moisture susceptibility of the asphalt mixture. Comparatively, ASA3 showed the best improvement in moisture susceptibility of asphalt binders and ASA2 was the worst. This result was consistent with the results of surface free energy and rolling bottle tests.

## 4. Conclusions

In this study, the main aims were to investigate the chemical composition of amide, phosphate ester, and fatty amine ASAs, and to analyze their anti-stripping mechanisms. The effect of ASA dosage on the physical and rheological properties of asphalt was discussed, and the anti-stripping performance of AMAs under dynamic water and high-temperature conditions was characterized. The following conclusions can be drawn:
The main elements of ASA1 are Fe, S and Cl. ASA2 is mainly composed of non-metallic elements such as P and S. ASA3 contains up to 96% Ca. Based on the results of infrared spectra, it is known that there is almost no chemical reaction between the AB and ASA, and the mechanism of the ASA can be classified as a physical modification.The optimum dosage of ASAs is 0.3%, as determined by the physical properties of asphalt. ASA1 exhibits the best basic physical properties, and ASA2 shows the best high-temperature deformation resistance, with a 46% increase compared to the base asphalt. ASA3 significantly improves the low-temperature performance of the asphalt, with a 5% improvement.ASAs increases the dispersion and polarity components in asphalt. Asphalt adhesion energy is improved and the stripping energy is reduced. ASA3 contains a large amount of metal cations and polar amine groups, which increased the polar component by 2.8 times and the ER value by 45%. The increase in these indexes is significantly higher than that of ASA1 and ASA2.Compared to other particle sizes of coarse aggregates, ASAs have the best modification effect on the moisture resistance of basalt with a particle size of 9.5–13.2 mm. Under different testing conditions, the adhesion properties of ASA1, ASA2, and ASA3 modifying the asphalt–aggregate interface are enhanced by 65%, 45%, and 70%, respectively.

The findings of this study provide a basis for investigating the mechanism of action of each component of the anti-stripping agent and its modified asphalt performance under multiple conditions, which can help engineers understand the effects of different types of anti-stripping agents on the physicochemical properties of asphalt and select the right type of anti-stripping agent according to the service requirements. However, the lack of anti-stripping agents and their associated adhesion models makes this paper incomplete in exploring the adhesion mechanism of anti-stripping agents in the asphalt–aggregate interface. In future research, it is recommended to conduct further studies on the effect of each component of the anti-stripping agent on the properties of asphalt and its coupling mechanism in combination with molecular dynamics simulations and other computer tools.

## Figures and Tables

**Figure 1 materials-16-04545-f001:**
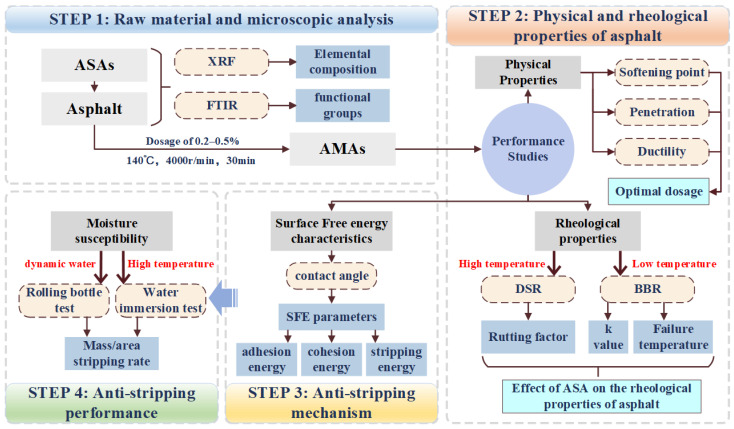
Research program.

**Figure 2 materials-16-04545-f002:**
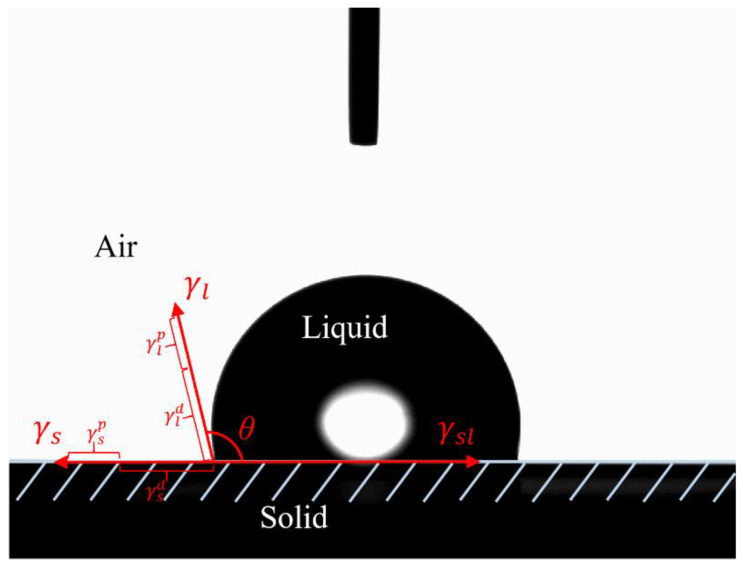
The calculation method of contact angle.

**Figure 3 materials-16-04545-f003:**
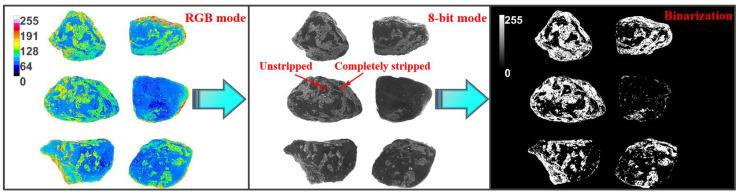
Image processing process for asphalt mixture.

**Figure 4 materials-16-04545-f004:**
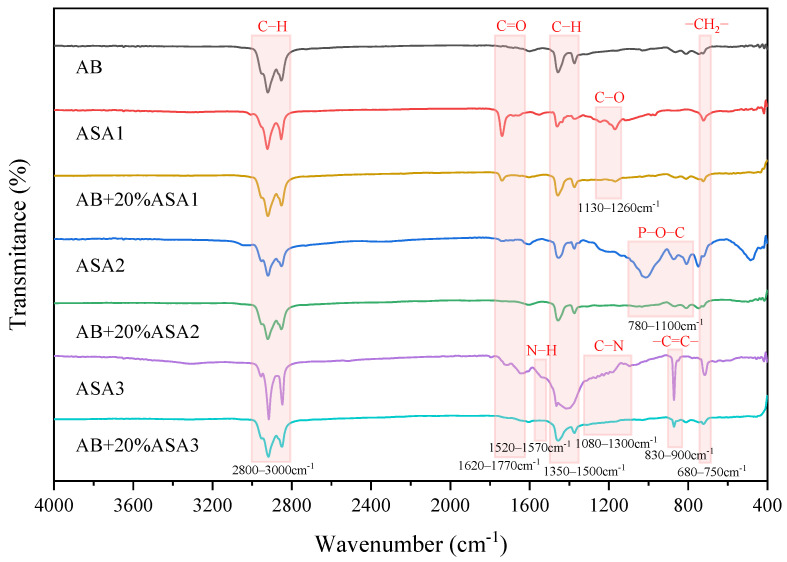
The FTIR spectra of ASAs, AB before and after adding ASAs.

**Figure 5 materials-16-04545-f005:**
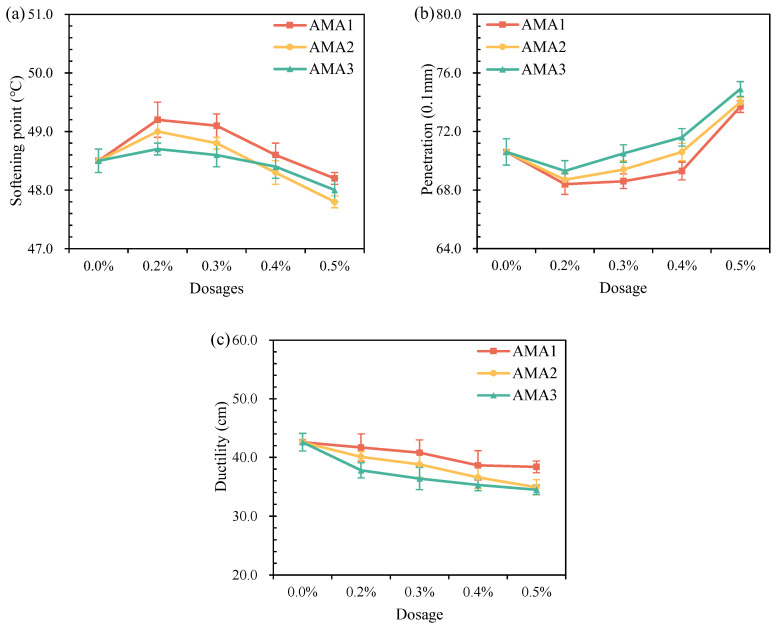
Effect of ASAs dosages on physical properties of AMAs: (**a**) 25 °C penetration; (**b**) softening point; (**c**) 10 °C ductility.

**Figure 6 materials-16-04545-f006:**
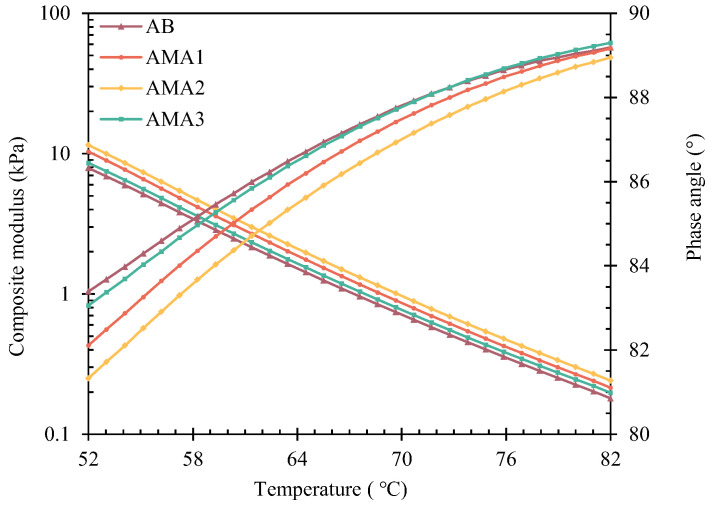
G* and δ of AB and AMAs.

**Figure 7 materials-16-04545-f007:**
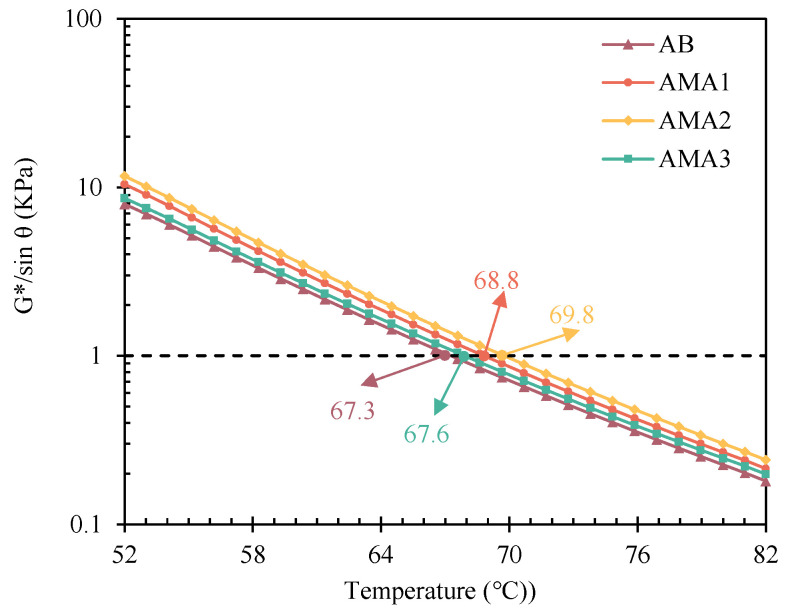
Rutting factor of AB and AMAs.

**Figure 8 materials-16-04545-f008:**
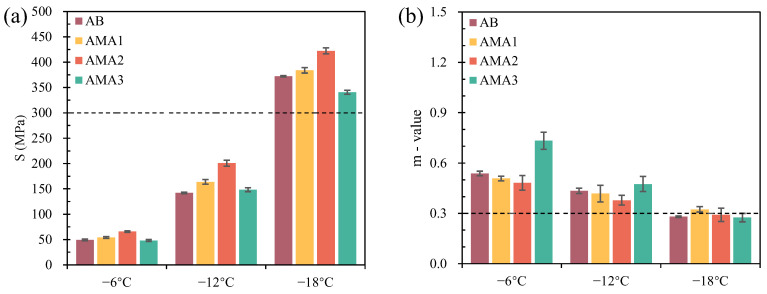
S and m values of AB and AMAs: (**a**) S; (**b**) m-value.

**Figure 9 materials-16-04545-f009:**
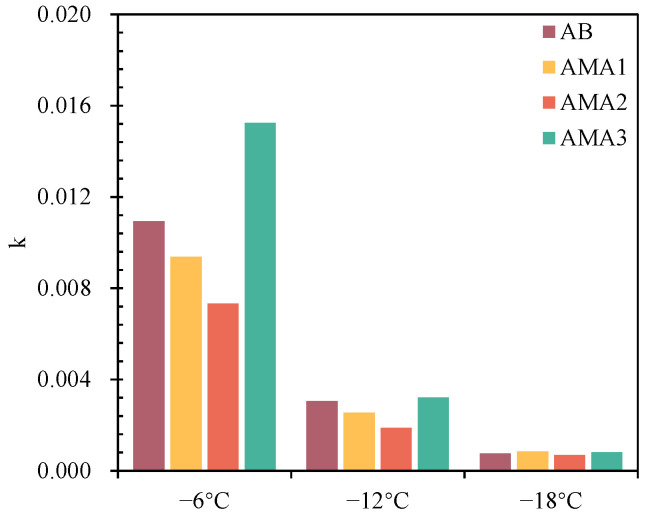
The k-value of AB and AMAs at low temperature.

**Figure 10 materials-16-04545-f010:**
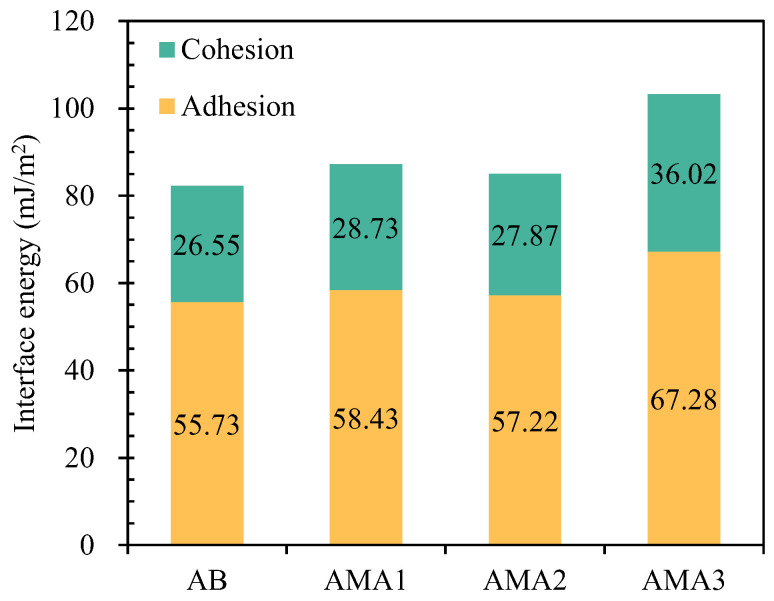
Adhesion and cohesion energy of AB and AMAs.

**Figure 11 materials-16-04545-f011:**
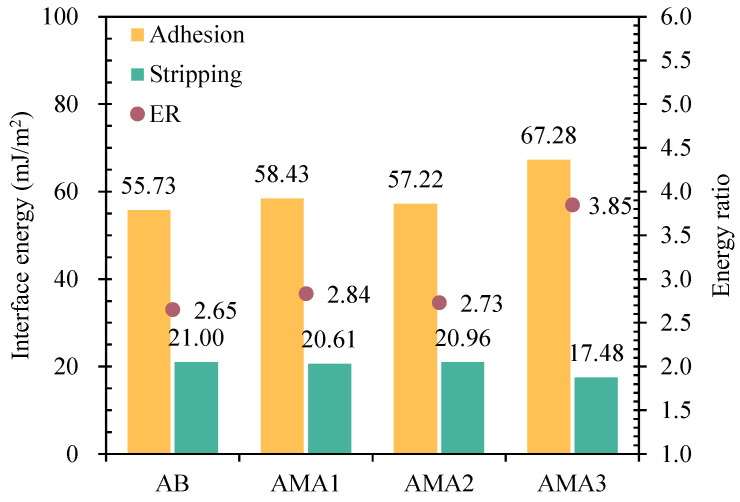
Moisture susceptibility indexes of AB and AMAs.

**Figure 12 materials-16-04545-f012:**
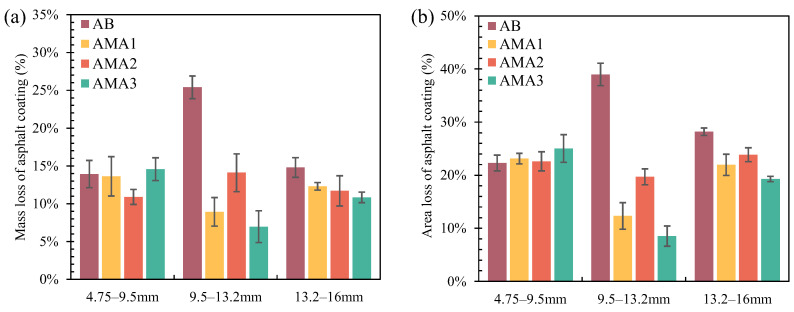
Asphalt stripping rate of different particle size mixes: (**a**) mass; (**b**) area.

**Figure 13 materials-16-04545-f013:**
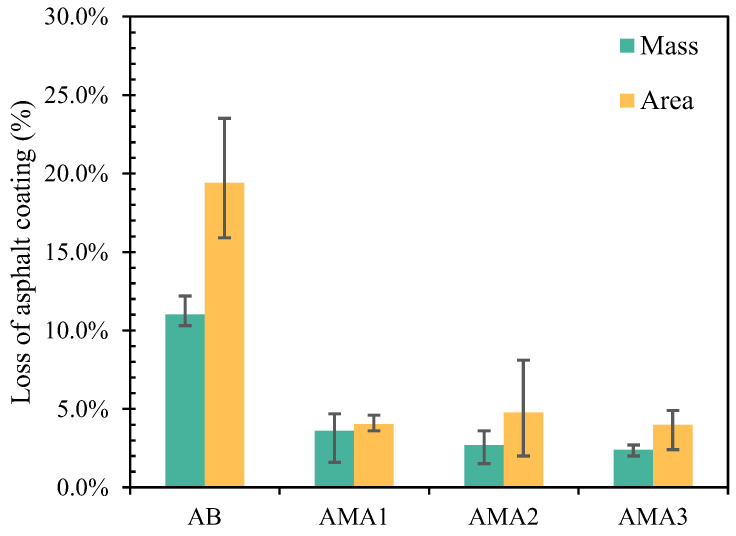
Mass and area loss of asphalt binder and AMAs.

**Table 1 materials-16-04545-t001:** Physical properties of AB.

Properties	Unit	Measured Values	Test Methods [19]
Penetration (25 °C, 100 g, 5 s)	0.1 mm	68.6	T0604
Ductility (10 °C)	cm	39.1	T0605
Softening point	°C	49.0	T0606
Solubility (trichloroethylene)	%	99.7	T0607

**Table 2 materials-16-04545-t002:** Physical properties of aggregates.

Properties	Particle Size	Basalt	Technical Indicators	Test Methods [20]
Apparent specific gravity (g∙cm^−3^)	13.2–16 mm	2.985	≥2.6	T0304
	9.5–13.2 mm	2.978	≥2.6	T0304
	4.75–9.5 mm	2.976	≥2.6	T0304
Water absorption (%)	13.2–16 mm	0.54	≤2.0	T0308
	9.5–13.2 mm	0.67	≤2.0	T0308
	4.75–9.5 mm	0.72	≤2.0	T0308
Los Angeles abrasion (%)		20.9	≤28	T0317
Crushed value (%)		16.1	≤26	T0316

**Table 3 materials-16-04545-t003:** Characteristics of anti-stripping agents.

Properties	Units	ASA1	ASA2	ASA3
Appearance	/	Brown viscous liquid	Dark-brown viscous liquid	Orange solids powder
Density (15 °C)	g·cm^−3^	1.05	0.95	1.00
Melting point	°C	0	−5	60
Viscosity (100 °C)	cSt	14	19	/
Flash point	°C	>220	>220	/

**Table 4 materials-16-04545-t004:** The SFE components of four test liquids (mJ/m^2^).

Test Liquids	Water	Glycerol	Formamide	Diiodomethane
$\gamma_{l}$	72.8	64.0	58.0	50.8
$\gamma_{l}^{d}$	21.8	34.0	39.0	50.8
$\gamma_{l}^{p}$	51.0	30.0	19.0	0.0

**Table 5 materials-16-04545-t005:** Chemical composition of aggregate and ASAs.

Component (%)	Basalt	ASA1	ASA2	ASA3
Na_2_O	4.98	11.900	3.888	0.286
Al_2_O_3_	16.11	4.753	2.656	0.473
SiO_2_	38.36	7.391	1.311	0.980
P_2_O_5_	0.352	6.501	27.161	0.046
SO_3_	0.395	19.433	57.812	0.251
Cl	0.27	15.805	5.364	0.085
Fe_2_O_3_	19.34	24.487	1.808	0.439
CaO	10.39	/	/	96.129
MgO	7.19	/	/	1.138
Cr_2_O_3_	0.085	7.947	/	/
Others	2.528	1.783	/	0.171

**Table 6 materials-16-04545-t006:** The special absorption bands of the ASAs.

ASA1		ASA2		ASA3	
Wavenumber /cm^−1^	Functional Group	Wavenumber /cm^−1^	Functional Group	Wavenumber /cm^−1^	Functional Group
1739	ν (C=O of carboxylic acid)	3028	ν (C−H of benzene ring)	3310	ν (N−H of fatty secondary amines)
1660	ν (C=O of tertiary amides)	1741	ν (C=O of aromatic esters)	1719	ν (C=O of carboxylic acid)
1551	ν (C=C of benzene ring)	1604	ν (C=C of benzene ring)	1643	ν (C=O of amide)
1247	ν (C−O of aromatic oxides)	1200	ν (P=O of phosphate ester)	1554	β (N−H of fatty secondary amines)
1170	ν_as_ (C−O−C of fatty ethers)	1013	ν_as_ (P−O−C of phosphate ester)	1315–1182	ν (C−N of aromatic ring)
		872	ν (Meta-substituted of benzene ring)	1095	ν (C−N of fatty secondary amines)
		808	ν_as_ (P−O−C of phosphate ester)	871	ν (Para-substituted of benzene ring)

**Table 7 materials-16-04545-t007:** Failure temperature of asphalt at low temperature.

Testing Samples	AB	AMA1	AMA2	AMA3
Fitting temperature of S (°C)	−16.2	−15.6	−14.4	−17.0
Fitting temperature of m-value (°C)	−17.5	−19.7	−17.3	−17.1
Failure temperature (°C)	−16.2	−15.6	−14.4	−17.0

**Table 8 materials-16-04545-t008:** Contact angle values of different samples at 20 °C.

Test Liquids	AB		ASA1 MAB		ASA2 MAB		ASA3 MAB	
	Avg. (°)	SD (°)	Avg. (°)	SD (°)	Avg. (°)	SD (°)	Avg. (°)	SD (°)
Water	104.0	0.5	105.0	0.1	104.4	0.6	93.4	0.3
Glycerol	102.2	0.3	101.8	0.5	101.1	0.4	92.6	0.3
Formamide	88.9	0.7	85.4	0.6	88.6	0.1	79.4	0.2

Note: Avg = average; and SD = standard deviation.

**Table 9 materials-16-04545-t009:** The SFE components of aggregate and AMAs.

Samples	Basalt	AB	AMA1	AMA2	AMA3
$\gamma_{s}^{d}$ (mJ/m^2^)	38.74	11.34	11.97	11.82	12.59
$\gamma_{s}^{p}$ (mJ/m^2^)	24.63	1.94	2.40	2.11	5.42
$\gamma_{s}$ (mJ/m^2^)	63.38	13.28	14.37	13.93	18.01

## Data Availability

The data presented in this study are available on request from the corresponding author.

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
