# Peer review of "Experimental Study on the Physicochemical Properties of Asphalt Modified by Different Anti-Stripping Agents and Their Moisture Susceptibility with Aggregates"

_materials, 2023, doi:10.3390/ma16134545_

Round 1

Reviewer 1 Report

Review on “Mechanism of Different Anti-Stripping Agents and Their Effects on Moisture Susceptibility of Asphalt-Aggregate Interfaces”

by Lu et al.

Manuscrip​t ID materials-2443533

A- General Comments

The paper in hand characterized the chemical composition of three types of anti-stripping agents (ASAs) and ASA-modified asphalts (AMAs), compared the physical properties of AMAs, determined the optimal dosage of different ASAs, and discussed the rheological properties of AMAs. The mechanism of action of the ASA was further discussed using the theory of surface free energy (SFE). Particularly, it was shown by authors that the phosphate ester ASA increased failure temperature for low-temperature rheology of asphalt by 11%. And the aliphatic amine ASA reduced the asphalt failure temperature to -17 °C, and the anti-deformation ability of asphalt was significantly improved

The topic of the paper is interesting, within the scope of the journal, and worthy of investigation. The originality of the work is acceptable and the study performed is adequate. However, the manuscript deserves a major revision. I suggest that authors take into account the comments and questions below before it can be accepted for publication in materials.

B- Detailed Comments and questions

Authors

Affiliations of authors should be provided.

Title

Details on how the study was performed can be helpful in the title. For instance, experimental study.

Abstract

1- The abstract should be shortened mainly by shortening the presentation of results at the end by keeping the most relevant ones.

2- It is not clear whether the originality resides in the suggestion of different Anti-Stripping Agents or the study of their effects on moisture susceptibility of Asphalt-Aggregate Interfaces or both. Please clarify.

Keywords

Keywords are ok.

1- Introduction

1- References relevant to materials should be added, if possible.

2- The originality of the work should be more highlighted at the end of the introduction especially with respect to the research gap in the field.

2- Materials and Methods

1- References to most of the equations should be provided;

2- Details on the uncertainty of the measurements should be provided;

3- Quality of Figure 2 should be enhanced;

4- This section lacks illustrative figures.

3- Results and discussion

1- There are a lot of interesting observations without deep analysis. More physical analysis is to be added to this section by shortening the quantity of results shown if needed;

2- The quality of all figures of this section can be enhanced.

4- Conclusion

The main outputs of the work in terms of applications should be highlighted.

5- References

References relevant to materials should be added, if possible.

The manuscript deserves an english language proofreading.

Author Response

Thank you for your review, the revised content is attached.

Reviewer 2 Report

The authors submitted "Mechanism of Different Anti-Stripping Agents and Their Effects on Moisture Susceptibility of Asphalt-Aggregate Interface" Here are my comments:

1. the authors should write the full name of ASAs in the highlight part

2. what is the chemical structures of ASAs and ASA-modified asphalts (AMAs)?

3. It is hard to review Figure 3; all samples profile looks the same.

4. Conclusion part is too long. please summarize this part.

5. How about the thermal stability of these materials and their morphologies?

Minor editing of English language required

Author Response

(The authors gave the same response as above.)

Reviewer 3 Report

This study assessed performance of using several antistripping additives on aggregate-binder interface. The manuscript is well-written and structured. In addition, the approach and experimental program as well as data analysis are sound leading to reliable conclusions. However, some revisions are needed before this manuscript can be accepted.

- The knowledge gap should be clarified clearer.

- authors should also describe the research limitations and contribution to the body of the literature.

- It needs to mention how many samples were conducted for each experiment and include the results variation to present the results.

- I would recommend the authors to add functional groups to Fig. 3. It will be more trackable for the readers. Please determine where exactly C=O stretching, CH2, CH3, and C-H bending occur and discuss

- Authors are suggested to add recommendations for further studies in conclusion section.

- the number of references is quite sufficient, however, it is advised to add more MDPI journals' papers as the number is only 7 papers out of 55. Here is my suggestion

1. https://doi.org/10.3390/su15043807 ( antistripping agent exhibited a better rutting resistance in a wet condition, also enhanced the moisture resistance of asphalt mixtures

2. https://doi.org/10.3390/coatings12121895 (anti-stripping agent not only improved the high-temperature deformation resistance of asphalt, but also improved the deficiency of rock asphalt’s low-temperature performance and effectively enhanced its performance)

3.  https://doi.org/10.3390/ma15030915  (asphalt modified by amine or organic polymers anti-stripping agent could significantly improve the adhesion between granite and asphalt)

also, trying to delete outdated references (before 2018)

Author Response

(The authors gave the same response as above.)

Round 2

Reviewer 1 Report

Thank you for taking into consideration my comments. The manuscript is now ready for publication.

Reviewer 2 Report

This manuscript could be accepted.

Minor editing of the English language required